# Association of Plasma Methylglyoxal Increase after Myocardial Infarction and the Left Ventricular Ejection Fraction

**DOI:** 10.3390/biomedicines10030605

**Published:** 2022-03-04

**Authors:** Stefan Heber, Paul M. Haller, Attila Kiss, Bernhard Jäger, Kurt Huber, Michael J. M. Fischer

**Affiliations:** 1Institute of Physiology, Center for Physiology and Pharmacology, Medical University of Vienna, 1090 Vienna, Austria; michael.jm.fischer@meduniwien.ac.at; 2Department of Cardiology, University Heart & Vascular Center Hamburg, University Medical Center Hamburg-Eppendorf, 20246 Hamburg, Germany; p.haller@uke.de; 3German Center for Cardiovascular Research (DZHK), Partner Site Hamburg/Kiel/Lübeck, 20151 Hamburg, Germany; 4Center for Biomedical Research and Translational Surgery, Ludwig Boltzmann Institute for Cardiovascular Research, Medical University of Vienna, 1090 Vienna, Austria; attila.kiss@meduniwien.ac.at; 53rd Department of Medicine, Cardiology and Intensive Care Medicine, Klinik Ottakring, 1016 Vienna, Austria; bernhard.jaeger@meduniwien.ac.at (B.J.); kurt.huber@meduniwien.ac.at (K.H.); 6Faculty of Medicine, Sigmund Freud University, 1020 Vienna, Austria

**Keywords:** acute myocardial infarction, methylglyoxal, cardiac function, remodeling

## Abstract

Background: Preclinical studies suggest that methylglyoxal (MG) increases within the myocardium upon acute myocardial infarction (AMI) and thereafter contributes to adverse postinfarct remodeling. The aims of this study were to test whether MG increases in plasma of humans after AMI and whether this increase is related to the left ventricular ejection fraction (LVEF). Methods: The plasma samples of 37 patients with ST elevation AMI undergoing primary percutaneous coronary intervention (pPCI) acquired in a previously conducted randomized controlled trial testing remote ischemic conditioning (RIC) were analyzed by means of high-performance liquid chromatography. Time courses of the variables were analyzed by means of mixed linear models. Multiple regression analyses served to explore the relationship between MG levels and the LVEF. Results: Compared to the MG levels upon admission due to AMI, the levels were increased 2.4-fold (95% CI, 1.6–3.6) 0.5 h after reperfusion facilitated by pPCI, 2.6-fold (1.7–4.0) after 24 h and largely returned to the baseline after 30 d (1.1-fold, 0.8–1.5). The magnitude of the MG increase was largely independent of that of cardiac necrosis markers. Overall, the highest MG values within 24 h after AMI were associated with the lowest LVEF after 4 d. While markers of myocardial necrosis and stretch quantified within the first 24 h explained 52% of the variance of the LVEF, MG explained additional 23% of the variance (*p* < 0.001). Conclusions: Considering these observational data, it is plausible that the preclinical finding of MG generation after AMI negatively affecting the LVEF also applies to humans. Inhibition of MG generation or MG scavenging might provide a novel therapeutic strategy to target post-AMI myocardial remodeling and dysfunction.

## 1. Introduction

### 1.1. Background

Despite significant improvements in the care of patients with acute myocardial infarction (AMI) in the recent decades, AMI remains associated with high morbidity and mortality [1]. Rapid restoration of blood flow has been proven beneficial; however, the resulting reperfusion of the ischemic myocardium acutely exacerbates tissue damage, a phenomenon known as ischemia-reperfusion injury [2,3]. Different interventions targeting ischemia-reperfusion injury have been tested in preclinical and clinical studies so far, although no intervention has yielded a convincing effect that would result in its implementation in the clinical routine, as reviewed previously [2]. One of these interventions is remote ischemic conditioning (RIC), whereby brief and repetitive ischemia and reperfusion cycles in a remote tissue or organ lead to tissue protection. In this regard, one can differentiate between pre-, per- and postconditioning, i.e., conditioning that takes place before, during or after the ischemia causing infarction [4]. Preclinical results have been promising [5,6], and preconditioning is impossible for the treatment of AMI in humans, but large clinical trials applying per- or postconditioning strategies have ultimately failed to show the cardioprotective effect of RIC in patients with ST elevation myocardial infarction [7,8].

After myocardial infarction, the main driver of morbidity and mortality is the resulting myocardial dysfunction, ultimately resulting in subsequent heart failure. An important determinant of the latter is the final infarct size [9]. However, myocardial dysfunction remains only partly explained by the loss of cardiomyocytes. As such, the concept of myocardial dysfunction following AMI involves several other processes. These include inflammatory ones that can be detected in the border zone of the infarcted myocardium, which largely contribute to adverse left ventricular remodeling resulting in cardiac dysfunction [10,11].

In this regard, advanced glycation end-products (AGEs) have been linked to worse outcomes after AMI. Specifically, an important AGE precursor methylglyoxal (MG) was shown in a preclinical study to causally affect myocardial dysfunction. Blackburn et al. [12] demonstrated that myocardial ischemia stimulates the production of MG within the myocardium. Overexpression of glyoxalase-1 (GLO1)—which constitutes the rate-limiting step in the detoxification of MG [13]—not only reduced the intramyocardial methylglyoxal levels, but also partially reversed the damage inflicted by infarction as measured with the left ventricular ejection fraction (LVEF). Notably, this functional difference occurred despite similar infarct sizes in both groups. Additionally, MG was shown as an independent predictor of the prognosis of patients with congestive heart failure [13]. This motivated us to investigate whether a negative association of MG levels after AMI and the subsequent LVEF exists in humans. In this case, translatability of the findings of Blackburn et al. [12] would seem plausible.

### 1.2. Objectives

Currently, it is unclear whether AMI in humans also leads to increased MG levels. Therefore, the primary objective of this study was to test whether MG levels increase over time in patients admitted to hospital due to AMI with ST elevation undergoing primary percutaneous coronary intervention (pPCI). To this end, the plasma samples previously acquired in the course of a randomized controlled trial investigating the effects of RIC on myocardial damage in patients with acute ST elevation AMI were analyzed. Thus, another exploratory objective was to test whether RIC affects MG levels, markers of myocardial damage and the myocardial function. The third objective was to investigate whether the negative impact of MG levels on the LVEF observed in mice [12] translated to a negative correlation of MG and the LVEF in patients.

## 2. Materials and Methods

### 2.1. Trial Design

The original study was a single-center, open-label, parallel-group randomized controlled trial. Patients with ST elevation AMI were randomized to RIC or sham in a 1:1 ratio. The associative research question addressed in the present report refers to observations of patients participating in the abovementioned trial. In this regard, the present analysis corresponds to an observational cohort study.

### 2.2. Participants, Setting and Recruitment

All the patients who presented with ST elevation AMI to the Third Medical Department (Cardiology) at Klinik Ottakring, Vienna, Austria, with intended interventional treatment by pPCI that gave written informed consent were included. We excluded patients if the symptom onset exceeded 8 h at the time of presentation to the emergency department, those with a regular intake of drugs affecting the K_ATP_ channel, patients with neurological disorders (i.e., diabetic neuropathy) and those in cardiogenic shock or other circumstances making informed consent impossible. In addition, four apparently healthy control subjects were included for assay development.

### 2.3. Interventions

The intervention was “remote ischemic conditioning” (RIC). It was carried out as previously described [14] using a blood pressure cuff applied to the left arm filled with a pressure of 200 mm Hg over 5 min. In case the patient’s systolic blood pressure exceeded 200 mm Hg, the cuff was inflated to exceed that pressure by 15 mm Hg. The RIC protocol consisted of four cycles of 5 min inflation each. In between, the pressure was released for 5 min to allow reperfusion. RIC was performed as soon as possible after admission and following informed consent but prior to and without delaying reperfusion of the culprit lesion. The patients in the sham control group were fitted with a blood pressure cuff for 40 min without inflating it.

### 2.4. Outcomes

The primary outcomes for the present analysis are the levels of MG in plasma, their association with the measured myocardial necrosis markers, RIC and the LVEF. Details can be found in the Section B.5.

### 2.5. Safety Considerations

A major concern could be that RIC delays the start of myocardial reperfusion. Therefore, RIC was started as soon as possible after the arrival of each patient to the hospital. In all those cases where the RIC protocol could not be fully performed before the start of reperfusion, the missing cycles were continued throughout the routine procedures and in case of very early reperfusion in the sense of “remote ischemic postconditioning”.

### 2.6. Sample Size

The present analysis includes all consecutive patients included from April 2016 to March 2018. No power calculation for the final sample considering the presented analyses was performed as this analysis is of exploratory character. The primary aim of the original randomized controlled trial was to confirm the previously reported benefit of RIC with respect to infarct size. White et al. [15] reported a reduction of infarct size from 24.5% in the control subjects to 18% in the RIC-treated ones. Based on their reported group difference and an assumed common standard deviation of 12%, it was calculated that 54 patients per group would be necessary to detect the relevant difference with a power of 80% accepting a type I error rate of 5%. As a dropout rate of 10% was expected, it was aimed for a total of 120 patients. The study was terminated after 37 of the intended 120 patients for the reasons provided below.

### 2.7. Randomization and Blinding

The allocation sequence was generated using a computer-based randomization tool generating 1:1 randomization in blocks of 8. The allocation sequence was available using serially numbered envelopes. The patients were informed that an additional intervention was offered as part of a clinical trial with a 50% chance of receiving this intervention, which might reduce damage to the heart. Due to the nature of the intervention, the patients and the physicians were not blinded regarding the group allocation. All the persons performing laboratory assessments were blinded to group allocation.

### 2.8. Methylglyoxal Measurements

Details on the methods, including Figure A1, Figure A2 and Figure A3, can be found in Appendix A.

### 2.9. Statistical Methods

Mixed linear models, partial correlation coefficients and multiple linear regression analyses are described in detail in Appendix B. The analyses were performed with IBM SPSS Statistics 27. They were not predefined in the study protocol, and no adjustment for multiplicity was performed; therefore, the results need to be interpreted accordingly. Only two-sided *p*-values are reported; *p*-values ≤ 0.05 were considered statistically significant.

## 3. Results

### 3.1. Participants

Between April 2016 and March 2018, 37 patients were randomized. In this period, several studies were published, which provide evidence that the beneficial effect of RIC is, at best, minimal [8]. Additionally, the recruitment of patients was far slower than expected, wherefore a decision was made to stop the study prematurely. The baseline patient characteristics were already published previously [14]. Plasma samples of 33 patients were available for MG measurements; the data of these patients are presented in this paper (Figure 1).

### 3.2. Myocardial Infarction Is Associated with a Transient Increase in Plasma Methylglyoxal Levels Independent of RIC

Compared to the MG values at 0 h, the levels were increased 2.4-fold (95% CI, 1.6–3.6, *p* < 0.001) after 0.5 h and 2.6-fold (1.7–4.0, *p* < 0.001) after 24 h. Thirty days after admission, MG levels did not significantly differ from the ones upon admission (1.1-fold, 0.8–1.5, *p* = 0.42). There was no evidence of an effect of RIC (interaction time * group *p* = 0.15, mixed linear model with time as a discrete variable, Figure 2A).

Exploration of the data using a mixed linear model with time as a continuous variable showed that they could be well-described by means of a cubic function (linear, quadratic and cubic terms *p* < 0.001 each, interactions with group *p* > 0.09 each), suggesting a RIC-independent nonlinear time course (Figure 2B,C, predicted vs. observed, Appendix B Figure A4A). The peaks of the estimated curves occur at 4 h and would correspond to a 6.7-fold (RIC: 6.1, sham: 7.2) increase compared to 0 h. 

### 3.3. Release of Enzymes Indicative of Myocardial Damage or Dysfunction Is Not Reduced by RIC after Myocardial Infarction

As markers of myocardial damage, CK-MB, total CK and TnI were repeatedly determined in plasma. To repeatedly assess myocardial dysfunction, NT-proBNP was quantified. Taken together, the following analyses are hardly consistent with a beneficial effect of RIC and rather point in the opposite direction, i.e., a detrimental effect.

CK-MB data (Figure 3A) were modelled using quartic polynomials (linear, quadratic, cubic, quartic terms *p* < 0.001 each; further increasing the order of the polynomial did not improve the fit; quintic term *p* = 0.067). Interactions of these terms with the factor group suggest different time courses for each group (Figure 3B, linear term * group *p* = 0.002, quadratic term * group *p* = 0.004, cubic term * group *p* = 0.009, quartic term * group *p* = 0.054). The estimated differences on a multiplicative scale (Figure 3C) suggest higher CK-MB levels in the RIC group compared to the sham group, with the difference most pronounced at around 2 h.

The time course of the total plasma CK levels were also well-described by a quartic polynomial (linear, quadratic, cubic, quartic terms *p* < 0.001 each); however, there was no evidence for group-specific parameters (all interaction terms *p* > 0.22, Figure 3D–F). Nevertheless, point estimates of the fold differences due to RIC are constantly well above one.

Concerning TnI levels (Figure 3G), the applied statistical model suggested a group-specific time course (Figure 3H, linear, quadratic terms *p* < 0.001 each, cubic *p* = 0.007, quartic *p* = 0.022, linear term * group *p* = 0.004, quadratic term * group *p* = 0.017, cubic term * group *p* = 0.038, quartic term * group *p* = 0.34). Similar to CK-MB, the estimated differences on a multiplicative scale (Figure 3I) show nearly threefold higher TnI levels compared to the sham treatment at around 2 h. Notably, the uncertainty of this estimate, quantified using the 95% confidence error band, needs to be considered. A sensitivity analysis showed no relevant impact on the estimates by the two highest outlying datapoints (Figure A5 in Appendix B).

Regarding NT-proBNP, the polynomial model suggested a group-specific time course as well (Figure 3J). In contrast to the abovementioned markers of myocardial damage, there was no evidence that a higher-order polynomial would fit the data better than a linear fit (linear and quadratic terms *p* = 0.005 and 0.24, Figure 3K). However, the slope of the lines differed between the groups (linear term * group *p* = 0.016), resulting in approximately 1.5-fold NT-proBNP levels increase towards the end of the day after MCI (Figure 3L).

### 3.4. Changes in MG Levels after AMI Are Only Weakly Dependent on Changes in CK, CK-MB, TnI or NT-proBNP, but Might Be Affected by Diabetes

As one would expect for different indices of the same underlying cause, i.e., CK-MB and TnI as markers of myocardial damage, they were strongly correlated within the patients (Figure 4A, partial r = 0.86, *p* < 0.001). TnI and NT-proBNP showed weak correlations with changes in methylglyoxal levels (NT-proBNP partial r = 0.22, *p* = 0.19, TnI partial r = 0.22, *p* = 0.14, Figure 4B,C). Myocardial damage markers CK-MB and CK were only moderately correlated with changes in methylglyoxal levels (CK-MB partial r = 0.41 *p* = 0.024, CK partial r = 0.45, *p* < 0.001, Figure 4D,E). When entered into a single multiple regression model, NT-proBNP, TnI, CK-MB and CK explained only 2.5% of the variance of methylglyoxal (*p* = 0.8). This suggests that the extent of myocardial damage cannot be the main determinant of the methylglyoxal level increase following AMI. As diabetes is associated with increased methylglyoxal levels, we explored whether the presence of diabetes type II might be an additional determinant. Although only two diabetic patients were among the included ones, these two patients indeed had comparably high methylglyoxal levels (Figure 4F), a difference that reached formal significance in the applied polynomial mixed linear model (interactions linear term * group *p* = 0.008 and quadratic term * group *p* = 0.008).

### 3.5. Association of Methylglyoxal Levels within 24 h after AMI with the Myocardial Function after 4 Days

To assess the relationships between myocardial damage, methylglyoxal levels and myocardial dysfunction, the area under the curve (AUC) of CK-MB, TnI, CK and NT-proBNP was calculated for 24 h after admission for each patient. The AUC values of CK-MB, TnI and CK were highly correlated with each other (Figure 5A, *p* < 0.001 each). Higher levels of the latter two markers for myocardial damage were weakly associated with higher NT-proBNP levels (Spearman ρ = 0.43, *p* = 0.01 and Spearman ρ = 0.41, *p* = 0.02), whereas no relevant correlation between CK-MB and NT-pro-BNP was observed (*p* = 0.16). Methylglyoxal showed no relevant association with any of the abovementioned variables. There were, however, moderate negative pairwise correlations between the LVEF and each of the biomarkers CK-MB, TnI, CK and NT-proBNP (*p* < 0.007 each). 

Next, a multivariable regression analysis was carried out to investigate whether methylglyoxal levels are associated with the LVEF independently of markers of myocardial damage and NT-proBNP. First, a model was built that predicts the LVEF (Figure 5B, Model 1) using CK, CK-MB, age, TnI and NT-proBNP. This model explained 52% of the variance of the LVEF (*p* < 0.001). Next, MG was added as a linear and quadratic term, the latter of which allows a nonlinear relationship between methylglyoxal and the LVEF. This Model 2 (Figure 5C) explained 23% more variance (R^2^ change *p* = 0.001) than Model 1, namely 75%. This means that methylglyoxal levels within the 24 h after AMI contained information regarding the LVEF measured approximately three days later in addition to the information contained in NT-proBNP, CK, CK-MB, TnI and age. The relationship between methylglyoxal and the LVEF was significantly curvilinear (logAUC(MG) linear and quadratic terms *p* < 0.001 each, Figure 5D). By plotting the predicted LVEF values against the range of observed AUC(MG) values, a curvilinear relationship can be appreciated, with the highest MG levels associated with the lowest LVEF. Additionally, there was no evidence that RIC affected the LVEF (Figure A6, Appendix B).

### 3.6. Harms

No serious adverse events related to placing the blood pressure cuff (and inflation in case of RIC) were observed. Overall, there was one death due severely impaired cardiac function following ST-elevation AMI in the RIC group.

## 4. Discussion

The main finding of this exploratory study is the observation that in patients admitted to a hospital due to ST elevation AMI who have undergone reperfusion by pPCI, MG levels in plasma rise substantially within a few hours and return to levels similar to the ones upon admission within 30 days. We speculate that AMI with subsequent reperfusion causes this transient elevation. It needs to be emphasized that based on our study design, it is not possible to deduce if AMI causes higher MG levels or if reperfusion by pPCI causes them. However, given the results in mice, which clearly show that AMI alone causes increased levels of MG-derived advanced glycation end-products within the myocardium [12], it is highly plausible that in humans AMI itself causes an increase in MG. However, a clear differentiation from a potential additional effect on MG levels due to reperfusion is not possible [16].

The second central finding is an association of higher MG levels within the first day of hospitalization with lower LVEF values after 4 days. In particular, an exploratory statistical analysis showed a nonlinear association between the methylglyoxal burden within the first 24 h after admission and the LVEF four days after AMI. Thereby, the highest MG levels seemed to be associated with the lowest LVEF values. This is in line with the previous preclinical data, showing that reducing the MG burden after AMI improves the LVEF [12]. Based on this preclinical observation, our study for the first time demonstrates a similar effect in humans with AMI undergoing reperfusion, suggesting a potential new target for treatment strategies. For instance, as soon as the exact mechanism of MG generation within the myocardium during AMI and following reperfusion is elucidated, one could try to interfere with it to improve the LVEF, e.g., by increasing the glyoxylase-1 activity. Alternatively, in case generation cannot be addressed, one could aim to scavenge MG, a concept that has been implemented experimentally [17]. It might also be possible that MG exerts its negative effects on the myocardial function by the formation of advanced glycation end-products. In this case, it might constitute an additional option to interfere with the interaction of MG with proteins or lipids.

Under the assumption that MG indeed reduces the LVEF after AMI, it seems relevant which factors determine the extent of MG increase thereafter. Moreover, the extent of MG increase seems to be independent of the measured surrogate markers for myocardial damage. Thus, other factors not involved in our models seem to alter MG. Among those, inflammation seems to be an important driver as previously shown in preclinical models [12]. Hence, although AMI might trigger the MG increase, its extent is determined by an unknown factor. A possible candidate factor could be diabetes, as this metabolic state is generally associated with higher MG levels [16]. Our observation that the two diabetic patients exhibited comparatively high MG increases after AMI compared to nondiabetic patients would fit this hypothesis; however, this needs to be investigated with an adequate sample size.

As the samples from an RCT investigating RIC in the context of AMI were analyzed, a RIC effect in MG or cardiac enzymes indicative of myocardial damage could also be assessed. In this regard, we found no evidence that RIC affects MG levels or markers of myocardial necrosis. Furthermore, our study did not reveal any significant benefit of RIC on these markers, nor did RIC have any significant influence on cardiac function thereafter. On the contrary, our analyses suggest a potential higher enzyme release in the RIC group that, based on the regression model, is most pronounced at 2 h after reperfusion. However, in sight of the smaller than expected sample size and the significant width of the confidence intervals, these results should be interpreted with caution and do not necessarily imply causality. Overall, previous studies and their pooled analysis showed a very small effect in favor of RIC on infarct size and cardiac function [8]. In line, the largest clinical trial conducted demonstrated a neutral effect with respect to occurrence of death or hospitalization for heart failure following AMI [7].

### Limitations and Generalizability

The results of this study seem plausible considering the published data, e.g., regarding the non-superiority of RIC compared to the control regarding myocardial damage or the MG increase after AMI and reperfusion that was already observed in preclinical studies. Nevertheless, several limitations need to be taken into consideration. Besides the effects of RIC, which were investigated in an experimental design, all other findings are of observational nature, i.e., no causal interpretation is possible. Consequently, it cannot be excluded that the MG increase is related not to AMI or reperfusion, but to an unknown confounder. Additionally, the low sample size adds some uncertainty, and validation of our results by independent researchers would be beneficial. Future studies should also include LVEF measurements at later timepoints after AMI. Another limitation concerns the multiplicity of statistical analyses. The polynomial modelling included several sequential decisions and thus might have resulted in false positive conclusions. Additionally, AUC calculations are based on the estimates derived from mixed linear models and thus are subject to some degree of uncertainty. Another limitation relates to the quantification of MG. In our view, the absolute concentration in plasma cannot be given due to matrix effects. For this purpose, a standard curve based on MG spiked, but otherwise an MG-free solution would be required. There is no method to remove MG from plasma without altering the matrix. However, the AUC of a given amount of MG obtained by HPLC heavily depends on the matrix, e.g., water, saline or plasma. The standard curve generated in H_2_O clearly did not reflect the DMQ/IS relationship to MG in plasma. Of note, this does not affect the conclusions of this study, which are based on relative MG concentration changes.

## 5. Conclusions

In conclusion, we found a temporal relationship between the occurrence of AMI treated with reperfusion and a subsequent increase in plasma MG. We also observed an association between the magnitude of the MG increase within the first day after the infarction with the left ventricular function four days later despite statistical adjustment for the known positive association of the established cardiac necrosis markers. Based on this observation, future studies investigating the causal relationship between MG and cardiac function in humans are warranted. Overall, MG might serve as a new target for the treatment of myocardial dysfunction, reperfusion injury and associated remodeling following AMI if confirmed in future randomized controlled trials.

## Figures and Tables

**Figure 1 biomedicines-10-00605-f001:**
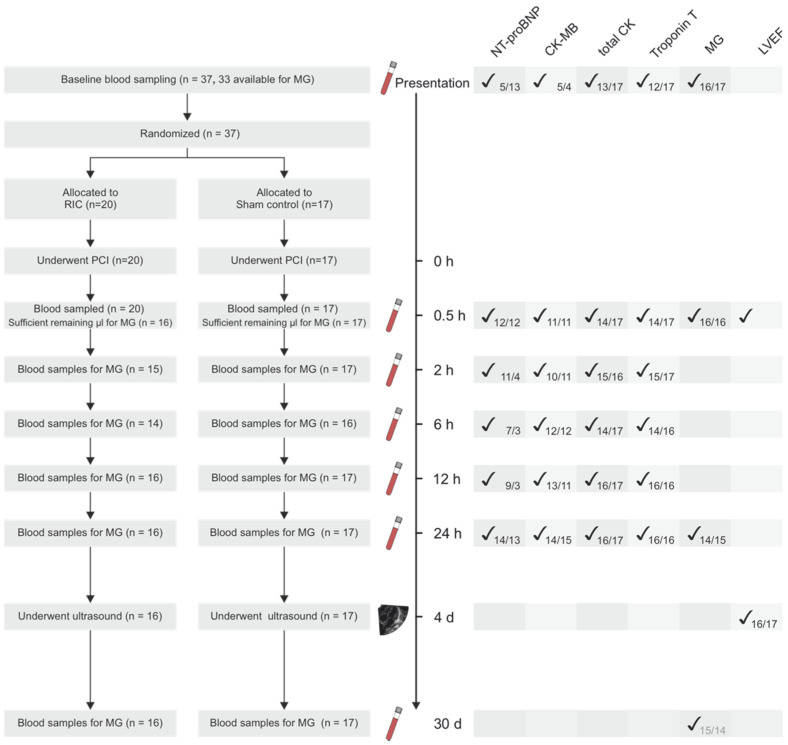
CONSORT flow diagram, analyzed parameters and sample size. The sample size given in the lower right corners next to the check marks indicate the sample size of the respective parameter at the timepoint. The number on the left of the slash represents the RIC patients, on the right—the sham patients.

**Figure 2 biomedicines-10-00605-f002:**
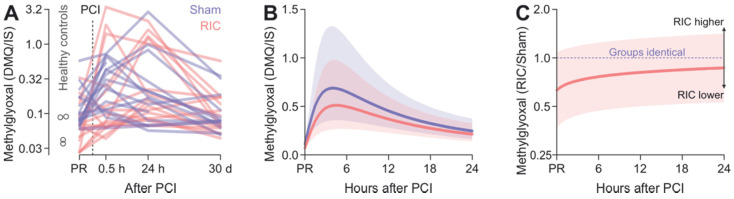
Time course of plasma MG levels after hospital admission due to myocardial infarction. (**A**) Individual time courses of plasma MG levels of the RIC and sham patients after presentation in the context of the plasma MG levels of four healthy control subjects as reference. Remote ischemic conditioning (RIC, red) or sham intervention (blue) were performed between 0 and 0.5 h. Logarithmic axis scaling allows visual discrimination between individual datapoints. (**B**) Estimated geometric mean time courses of methylglyoxal within the period until 24 h after admission. Error bands are 95% confidence intervals. (**C**) Estimated fold difference of MG levels between the RIC patients (red) and the sham patients (blue). Abbreviations: MG—methylglyoxal, RIC—remote ischemic conditioning, PR—presentation.

**Figure 3 biomedicines-10-00605-f003:**
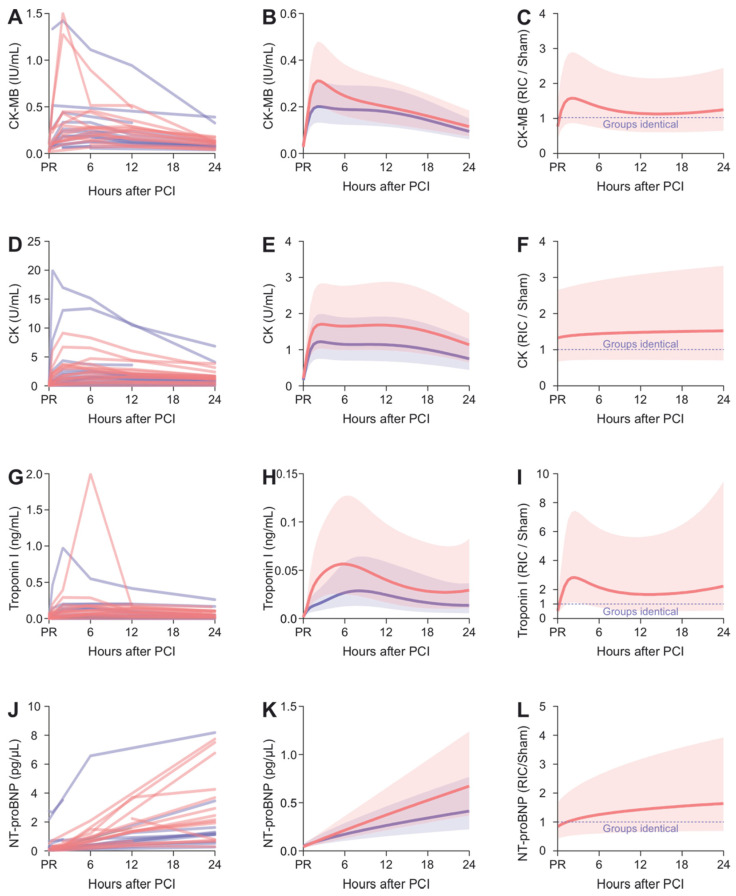
Biomarker levels in plasma after admission to the hospital due to myocardial infarction. The left panels show the raw values, with red lines indicating RIC and blue lines indicating sham control. The middle panels show the geometric means with 95% confidence bands estimated using mixed linear models. The right panels show the estimated fold difference of the RIC group compared to the control group with 95% confidence bands. Panels: (**A**–**C**) CK-MB, (**D**–**F**) CK, (**G**–**I**) TnI and (**J**–**L**) NT-proBNP.

**Figure 4 biomedicines-10-00605-f004:**
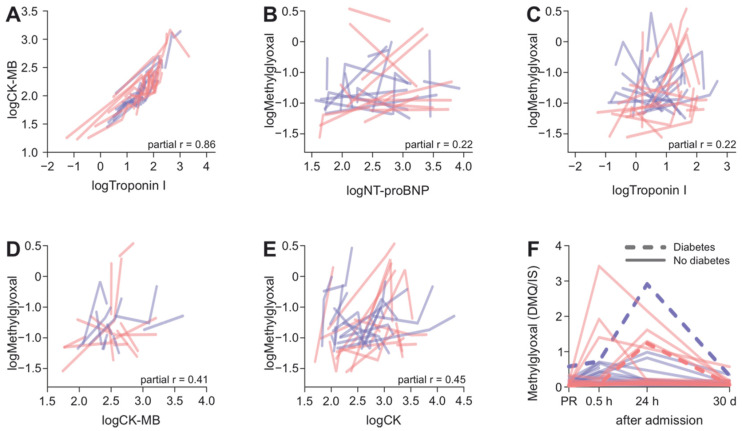
Correlations between myocardial damage and methylglyoxal. (**A**) Relationship between changes in TnI levels and changes in CK-MB. Each line represents measurements taken at different timepoints from a patient. Red indicates RIC, blue—sham. Correlation of changes can be appreciated by approximately parallel lines. (**B**) Correlations between intraindividual changes in methylglyoxal levels with changes in NT-proBNP, (**C**), TnI, (**D**) CK-MB and (**E**) CK. (**F**) Methylglyoxal levels of two patients with diagnosed type 2 diabetes (dotted lines) in the context of respective levels of all the nondiabetic patients.

**Figure 5 biomedicines-10-00605-f005:**
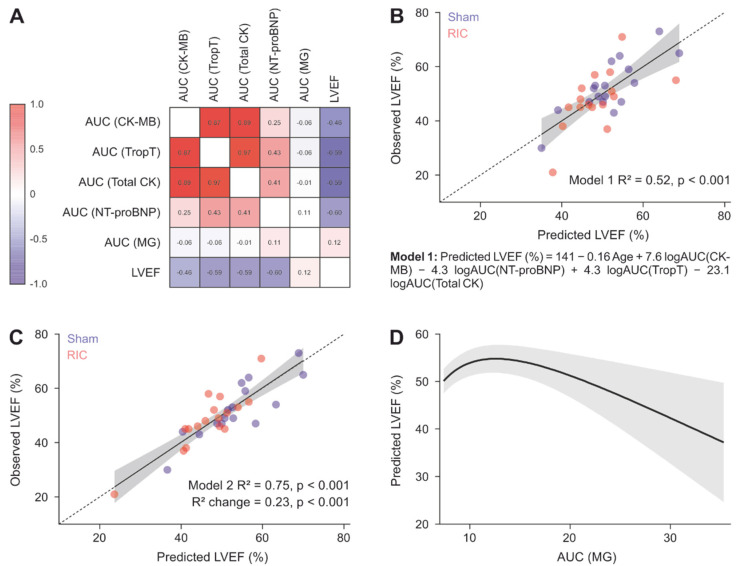
Methylglyoxal levels partially explain the myocardial function after myocardial infarction. (**A**) Changes in the parameters except for the LVEF were assessed by their area under the curve within the first 24 h. Heatmap of the Spearman correlation matrix including AUCs of enzymes in plasma indicating myocardial damage, AUCs of NT-proBNP, AUCs of methylglyoxal and the left ventricular ejection fraction. (**B**) LVEF values predicted without methylglyoxal. Entering each patient’s values into the formula results in their predicted LVEF. The dotted line represents the perfect prediction. (**C**) Plot analogous to B, with inclusion of MG as a linear and quadratic predictor. (**D**) Relationship between AUC(MG) and the LVEF. The covariates other than MG were kept constant at their means. The AUC(MG) values in the observed range were entered into Model 2.

## Data Availability

Data are available upon reasonable request after contacting the corresponding author.

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
