# Peer review of "Association of Plasma Methylglyoxal Increase after Myocardial Infarction and the Left Ventricular Ejection Fraction"

_biomedicines, 2022, doi:10.3390/biomedicines10030605_

Round 1

Reviewer 1 Report

This study demonstrated that methylglyoxal (MG) transiently increased after percutaneous coronary intervention (PCI). The increase in MG might have a negative effect on left ventricular function. On the other hand, remote ischemic conditioning (RIC) was not shown to be useful in treating myocardial damage caused by reperfusion-stress during PCI. The results of this study, which firstly reveled the changes of MG after PCI, are significant in cardiovascular research. Although this manuscript has important information and is well written in most areas, there are some points that could be improved. These are given below.

  1. The underlying conditions of the patients in the previous study by the authors (Biomedicines 2020, 8, 218) had showed that there were more hypertensive patients in the RIC group than in the Sham group. Since the relationship between MG and hypertension has been reported in many studies, this difference may have been an important confounding factor for the experimental results. The authors should discuss about it.

  1. In the third paragraph of the discussion, the authors stated that “AMI increased MG”, but “PCI increased MG” might be correct. This is because the authors did not measure MG after AMI without PCI.

  1. One of the main stress by PCI-induced reperfusion might be an oxidative stress. MG is produced via the glycolytic system. Is it known whether oxidative stress activates this pathway and increases MG production? The authors should discuss more about the PCI-induced MG production.

  1. MG had a transient increase followed by a decrease. Was it due to the metabolism of MG into AGEs or a decrease in MG production? If AGEs are produced, do they have any adverse effect on the myocardium?

  1. In the second paragraph of the discussion, the possibility that MG control may be useful as a therapeutic target is discussed. In the relationship between MG (AUC) and LVEF in Figure 5D, there are areas (~10-15) where LVEF increases when MG concentration is low, suggesting that it is not necessarily better to suppress MG production. It may be that comprehensive management, including not only simple MG reduction but also control of the production of AGEs, a metabolic product, is necessary.

  1. Based on the results of this study, it is difficult to state clearly whether MG is important as a prognostic marker or a therapeutic target after PCI. Therefore, it would be better to weaken the assertion of the conclusion.

Author Response

We would like to thank you for the evaluation of our manuscript. Please find  our response to your comments in the attachment, including the information on what has been changed in the manuscript based on your suggestions.

Reviewer 2 Report

In the present study authors reported that increased methylglyoxal (MG) after acute myocardial infarction (AMI) negatively effects left ventricular ejection fraction (LVEF). This is an interesting work but there are some concerns which needs to be addressed.

Comments:

  1. In the present study, authors have used total n = 37 participants (n = 20 RIC and n= 16 Sham group). To conclude that there is an association between MG, AMI and LVEF authors should increase the n numbers.
  2. Do authors find any variations among males and female group? Also, does age has some effect on association of MG and AMI?
  3. Why have authors measured LVEF only at day 4 and not at day 30?
  4. Authors have mentioned that primary outcome of the study is association of MG with infarct size. Authors are advised to include measured infarct size in the present study to highlight the strong association of MG and AMI
  5. To validate the findings of the study authors should test the hypothesis that MG intervention can improve LVEF in an animal model.
  6. How authors are making sure that MG detected is only because of AMI and not by patients’ other disease conditions.

Author Response

(The authors gave the same response as above.)

Reviewer 3 Report

Very interesting study with good intentions for clinical practice. I have no particular comments. 

Author Response

We thank the reviewer for their encouraging words!

Round 2

Reviewer 2 Report

Authors have addressed the concerns and have included them in the revised manuscript. This manuscript can be published in the present form.